# Layout Evaluation of New Energy Vehicle Charging Stations: A Perspective Using the Complex Network Robustness Theory

**Peipei Zhang** **, Juan Chen \*, Lilan Tu and Longteng Yin**

College of Science, Wuhan University of Science and Technology, Wuhan 430065, China; zhangpeipei@wust.edu.cn (P.Z.); tulilan@wust.edu.cn (L.T.); zhang411521@outlook.com (L.Y.)
\* Correspondence: jc1204@126.com

**Abstract:** At present, the new energy vehicle industry is developing rapidly, but the relative lag in the development of its supporting infrastructure, especially charging stations, has become a bottleneck that restricts the development of the electric vehicle industry. In this paper, we propose a model for constructing a network of new energy vehicle charging facilities based on complex network theory and analyze the operation and the rationality of the layout of the new energy vehicle (NEV) charging stations in Wuhan and Hangzhou, respectively. The results show that the current layout of new energy vehicle charging stations in the city is relatively reasonable, but the allocation of charging pile resources is unreasonable. Our results of the virtual charging station network constructed by adding new charging station nodes show that the change in network structure helps to enhance the performance of the charging station system.

**Keywords:** charging stations layout evaluation; complex networks; network robustness

## 1. Introduction

The world's crude oil resources are expected to be exhausted within 100 years according to the current consumption rate. Transportation is the major oil-consuming industry of the entire society. Developing new energies is extremely urgent. At present, the global energy and environment system is facing huge challenges, and automobiles, as major oil consumption and greenhouse gas emitters, need to be completely changed. In recent years, the United States [1], Japan [2], Germany [3], South Korea [4], and other countries have formulated industrial policies to support the development of NEVs (new energy vehicles) in terms of strategic planning, research and development innovation, promotion, and application, and intelligent network connection.

A key element in popularizing NEVs is the planning and construction of NEV charging stations [5]. At present, countries all over the world are speeding up the process of construction of charging stations to promote the development of electric vehicles. However, the typical problem is that the development of new energy vehicles is not harmonized with the charging infrastructure, and the construction of charging facilities is unreasonable, which severely hinders the progress of NEVs. During the expansion of the electric vehicle industry, there is a widespread problem of paying attention only to cars but not to the construction of charging infrastructure. On the one hand, the growth rate of electric vehicles in some areas is fast, but the scale of charging facility construction is seriously insufficient; on the other hand, there is also an unreasonable layout and uneven distribution of the charging facilities, resulting in the inadequate utilization rate of charging facilities. The existing charging and battery technology is in the primary stage, resulting in a long recharging time, leading to strong "travel anxiety" among electric vehicle users.

Consumers' choice to purchase an electric vehicle depends on the convenience of public charging facilities. The scientific and reasonable location of the charging infrastructure and the scale of charging stations not only affects the service quality, operational

efficiency, and safety of charging stations but also directly relate to the convenience of electric vehicle users and the effectiveness of resource allocation, which in turn affects the scale development and application of electric vehicles.

Therefore, the issue of siting and layout planning of NEVs charging stations is gradually becoming the focus of scientific research. Some researchers have done some work on the factors affecting the siting of charging stations. The location and layout of charging stations should take into account the construction cost of charging stations, charging station benefits, user charging convenience, and other objectives, as well as the local grid load level, electricity prices, the number of electric vehicles, government policies, and other factors as constraints, which form a multi-objective optimization problem. Based on the difference in evaluation methods, Wang and Zhang introduced data envelopment analysis (DEA) as an effective to deal with the risk due to uncertainty and help decision-makers to obtain the optimal charging station planning solution [6]. Guo and Zhao applied a multi-criteria decision-making (MCDM) method based on fuzzy TOPSIS to optimize the location of electric vehicle charging stations [7]. Hayajneh and Zhang developed a simulation model based on the idea of co-evolution to study the interaction between charging station configuration and EV users and then used this modeling framework to evaluate the performance of planned charging stations for EVs. The modeling framework was then used to evaluate the performance of planned charging stations to provide services to EVs [8]. These methods can be used to evaluate the rationality of the charging station layout from a macro perspective. However, factor selection is influenced by the researchers' knowledge structure, cognitive level, and emotional factors, and site selection still contains some subjectivity, which weakens the comparability of quantitative factors.

The key to enhancing the efficiency of using charging facilities is to magnify the allocation of resources and establish a robust charging network. Yi analyzed the dynamic characteristics of new energy vehicle charging behavior, established a mathematical model of new energy vehicle charging service system queuing, and studied the impact of different configurations of charging facilities on the grid load to minimize the comprehensive cost [9]. The results show that if the charging facilities are reasonably configured, it can not only ensure the comprehensive satisfaction of the service system but also improve the utilization rate of facilities and increase the grid load rate. Zhou provided a practical model for the location decision of photovoltaic power charging stations which combines geographic information system (GIS) with multi-criteria decision-making (MCDM) methods, GIS has an important role to integrate spatially referenced data as a piece of the problem-solving environment [10]. Davidov used the set model to represent road networks and electric vehicle curves to analyze the influence of roads on charging infrastructure layout factors [11]. Dharmakeerthi proposed that charging preferences of new energy vehicle users and power system construction options affect new energy vehicle charging infrastructure planning and construction [12].

The charging facility network as a system involves several elements, and the correlation between the elements and the overall topology of the system has attracted extensive attention. Fang established a complex network-based evolutionary game model considering charging facility policy incentives and new energy vehicle user preferences and pointed out that charging prices and the popularity of new energy vehicles influence the future development of charging infrastructure [13]. Wang Wentao and Xu Xianyuan used network structure characteristic indexes to evaluate the layout of the charging facilities in China and concluded that the layout of existing charging facilities in cities could not be interconnected within effective charging areas [14].

In comparison with existing research, scholars have primarily focused on optimizing the layout of charging facilities and improving the operation pattern. However, the systemic role of urban new energy vehicle charging facilities as a system is ignored, as well as the direct interaction of the charging stations within the system. This paper applies the complex network theory to construct a charging facility network and evaluate the rationality of the layout of new energy vehicle charging stations in cities. We find that: (1) the

robustness is poor both in WNSN and HNSN. The NEV charging stations in Hangzhou are distributed evenly, but the whole charging station network is inefficient; (2) the topological characteristics and the number of charging piles of the charging station are considered comprehensively, which can better maintain the stability of the network; (3) the new station nodes can significantly improve the connectivity of the network and can reduce the probability of users not being able to charge continuously on the path within the region. The innovations and contributions of this paper are these:

1.  The research used complex network and network robustness theory to open up a new way of thinking about the operation mode of electric vehicle charging facilities and innovates the development path of the new energy business. It emphasizes the topological characteristics of the charging station network in the study of the rationality of the layout of charging facilities and highlights the connection between stations in the urban charging station network. Therefore, this paper analyzes data of charging stations in Wuhan and Hangzhou and builds a network of NEV charging stations in the two cities to study the dynamic characteristics of network efficiency and connectivity changes under malicious and random attacks, and to analyze the efficiency and the rationality of the layout of urban charging stations.
2.  The minimum eigenvalue of the Laplacian matrix after node deletion can reflect the importance of the node in the original network. This paper uses it for the first time as an attack indicator to analyze the robustness of the network and finds that the indicator is more effective in attacks on scale-free networks, but not significant in attacks on small-world networks with uniform degree distribution.
3.  In addition, this paper also constructs a virtual HNSN by adding 19 new station nodes to the real HNSN to verify whether the addition of new nodes improves the efficiency of the network and optimizes the layout of charging stations. The results show that the addition of new station nodes can significantly improve the connectivity of the network, enhance its resilience to betweenness centricity attacks, and can reduce the probability of users not being able to charge continuously on the path within the region.

## 2. Methodology

### 2.1. An Evaluation Procedure

In this paper, we use the robustness of the network to analyze the layout of urban NEV charging stations. The procedure for evaluating robustness is proposed and presented in Figure 1, it involves the following steps: (1) establish a network; (2) classify nodes according to their importance; (3) calculate robustness; (4) improve robustness. More precisely, the data for the first stage are based on the mobile client named Chongdianba. Moreover, the data only include open charging stations in urban areas, excluding private charging stations and suburban charging stations. In the first step, the key procedure involves processing the location of charging stations and calculating the distance between charging stations. The third stage: robustness is calculated based on the degradation of network performance. In the final stage, a virtual network was built to test whether its robustness was reinforced.

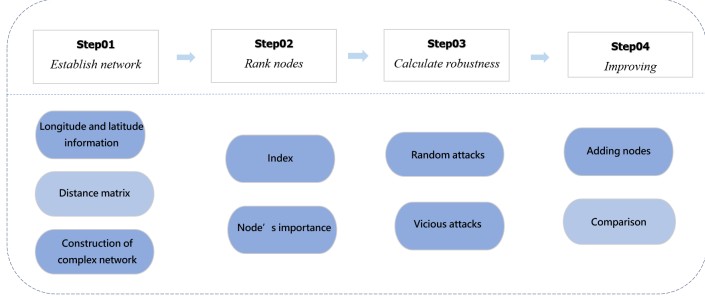

**Figure 1.** The robustness evaluation procedure.

*2.2. NEV Charging Station Network Construction*

2.2.1. Construction of the WNSN and HNSN

Figure 2 shows the distribution of NEV charging piles in the urban area of Hangzhou and Wuhan. City administrative area division data cited from the national geographic information resources directory service system (www.webmap.cn accessed on 6 June 2022). In Figure 2, the color close to red indicates that the number of charging piles distributed in the area is greater, and the closer to blue, the fewer charging piles distributed in the area. In Figure 2a,b, it is not hard to see that the distribution of charge stacks in the urban area of Wuhan and Hangzhou is mainly concentrated in the main urban areas of the two cities, such as Wuchang District, Hongshan District, and Jianghan District in Wuhan, Gongshu District, Binjiang District, Shangcheng District, and Xihu District in Hangzhou. These areas are either traditional central business districts, concentrated education areas, or traditional tourist areas, all of which are characterized by high population density and convenient transportation. Even in urban areas with high traffic volumes, the number of charging piles is still low, which means that many car owners have to wait in the charging queue, resulting in inefficient charging. It reflects that the current layout of new energy vehicle charging stations in the city is reasonable, but the allocation of charging pile resources is unreasonable, resulting in the current situation of "difficulty charging" for NEV owners. Comparing Figure 2a,b, we can see that compared with the distribution of charging stations in Wuhan, the distribution of new energy vehicle charging stations in Hangzhou is more concentrated, but the number of middle-scale charging stations in Wuhan is larger than that in Hangzhou.

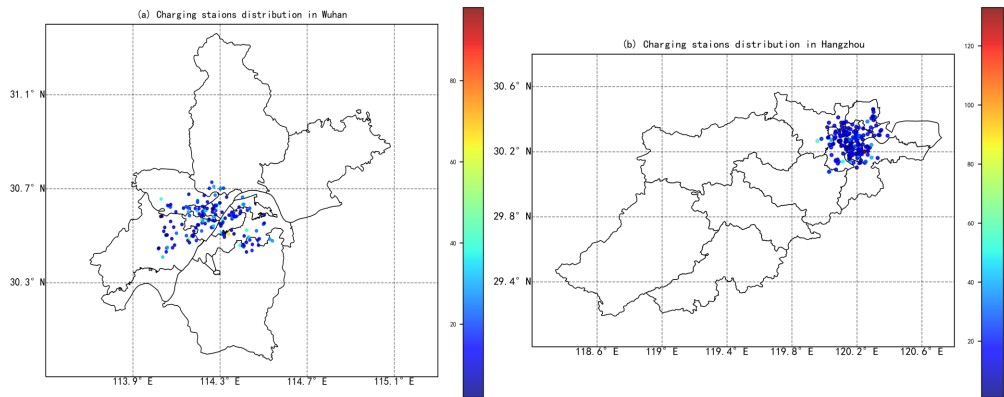

**Figure 2.** Distribution of NEV charging stations in urban areas. (**a**) Distribution of NEV charging stations in urban areas of Wuhan. (**b**) Distribution of NEV charging stations in urban areas of Hangzhou.

A complex network is a unified whole with numerous nodes and edges. This paper selects 157 and 185 charging stations in the urban area of Wuhan and Hangzhou respectively, regards the charging stations as the nodes of the network, and forms the connection between the charging facilities whose navigation distance between the nodes is less than 10 km. The NEV charging stations complex network in Hangzhou is constituted of 185 nodes and 3229 edges. The NEV charging stations complex network in Wuhan is constituted of 157 nodes and 1865 edges. Figure 3 illustrates the actual network of new charging stations in the urban area of Wuhan and Hangzhou, where nodes of different colors represent that they belong to different communities. We use the classic Louvain algorithm to perform community detection on WNSN and HNSN and find that they both have five communities. Nodes in the same community are more closely connected. Compared with small-scale communities, the three large-scale communities are more closely connected. This indicates that recharge resources are more plentiful within the community and across larger communities.

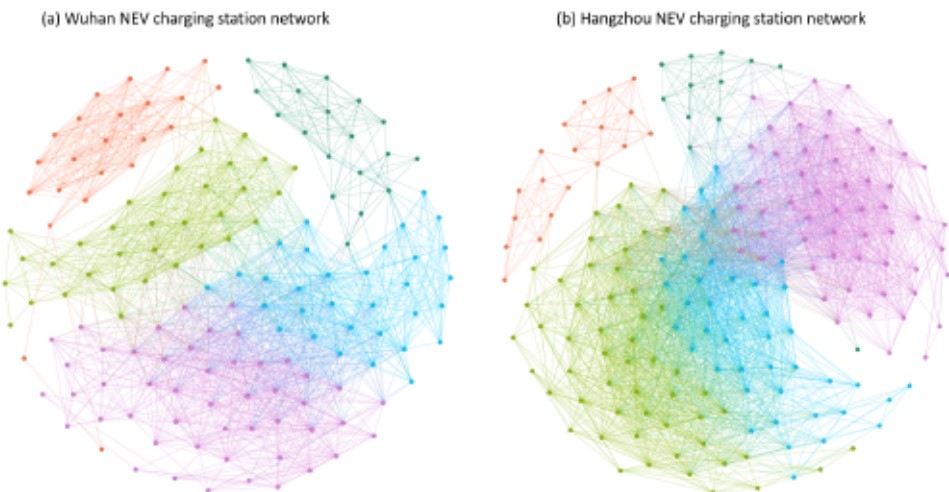

**Figure 3.** NEV charging station network in urban areas. (**a**) WNSN. (**b**) HNSN. The colors of the node indicate different communities.

Complex interactions, relationships, and interdependencies exist between the nodes. The complex network evaluation is carried out through the following steps:

Designing the complex network. We denote a complex network as $G = (V, E, A)$, and $V = \{v_i,\ i = 1, 2, \cdots, n\}$ is the set of nodes, where each node represents a NEV charging station; $E = \{e_j,\ j = 1, 2, \cdots, m\}$ is the set of edges between nodes $v_i$ and $v_j$. $A = (a_{ij})$ denotes the adjacency matrix, where $a_{ij}$ represents whether an edge exists between nodes $v_i$ and $v_j$, $a_{ij} = 1$ represents an edge whereas $a_{ij} = 0$ represents no edge, which is decided by

$$a_{ij} = \begin{cases} 1, & \text{if } g_{ij} \leq 10 \text{ km} \\ 0, & \text{otherwise} \end{cases} \tag{1}$$

where $g_{ij}$ is the navigation distance of nodes $v_i$ and $v_j$.

- Node degree:

The degree of a node measures the number of other nodes that connect with the node. Obviously, the greater the degree value of a node, the more crucial the node is in the network [15]. In a charging facility network, this means that the charging station is conveniently located and has a large number of charging facilities in close proximity. In the network, the number of neighboring edges of a node is called the degree of the node $v_i$. The node degree of node $v_i$, represented by $k_i$, is given by Equation (2).

$$k_i = \sum_{j=1}^{n} a_{ij} \tag{2}$$

2.2.2. Topological Properties of the Network

- Node betweenness centrality:

Betweenness centrality measures are aimed at summarizing the extent to which a node is located "between" other pairs of nodes. These centralities are based upon the perspective that "import" relates to where a node is located concerning the paths in the network graph. The higher the betweenness centrality of a node, the greater the contribution of that node to the connectivity of the network nodes [16]. In the case of the HNSN, the loss of high betweenness centrality of charging station nodes increases the rate at which users are

unable to recharge continuously on the routes within the region. The node betweenness centrality of the node $v_i$, represented by $BC_i$, is given by Equation (3)

$$BC_i = \sum_{j \neq k}^{n} \frac{N_{jk(i)}}{N_{jk}} \tag{3}$$

where $BC_i$ is the value of the betweenness of the node $v_i$, $N_{jk(i)}$ is the number of the shortest path across the node $v_i$ between the node $v_j$ and node $v_k$, and $N_{jk}$ is the number of the shortest path between the node $v_j$ and node $v_k$.

- The minimum eigenvalues of the deleted Laplacian matrix:

Some recent studies have shown that, according to the synchronization criterion of the pinning control, the synchronization of the pinning control of the network depends on the minimum eigenvalue of the grounded Laplacian matrix obtained by deleting the row and columns corresponding to the pined nodes from the Laplacian matrix of the network [17,18]. Calculating the minimum eigenvalues $\lambda_1(L_{N-l})$ of the deleted matrix can analyze the importance of the deleted nodes of the original matrix, the minimum eigenvalue is bigger, the node removed is more important, $l$ is the number of nodes that we will remove, $L_{N-l}$ is the grounded Laplacian matrix obtained by deleting the rows and columns corresponding to the deleted nodes from the Laplacian matrix of the network.

- Robustness analysis:

To analyze the characteristics of the robustness of HNSN, we use two measures, the connectivity of a network and the efficiency of the network, to quantify the robustness of networks against the number of removed nodes. The higher the connectivity, the more connected the charging facilities in the HNSN are, and the more charging resources are available to users when they need to recharge their NEVs in the region.

The efficiency of the network is calculated as the inverse of the shortest path length. The calculation of the global efficiency is given by Equation (4)

$$\mu = \frac{1}{n(n-1)} \sum_{i \neq j}^{n} \frac{1}{SL_{ij}} \tag{4}$$

where $\mu$ is the global efficiency of the complex network, $n$ is the number of the nodes in the network, and $SL_{ij}$ is the length of the shortest path between the node $v_i$ and node $v_j$.

Given an order of node attacks on a network with $n$ nodes, the robustness value of the network is defined as Equation (5). In a complex network, the network consisting of mutually accessible nodes is called a connected subgraph, and the one with the highest number of nodes is called the maximum connected subgraph of the network. The connectivity is the ratio between the number of nodes in the maximum connected subgraph and the total number of nodes in the network. If the maximum connectivity ratio of a network is 1, it means that all nodes of the network are reachable by each other.

$$G = \frac{S_i}{n} \tag{5}$$

where $S(i)$ is the relative giant connected component size after attacking the $i$th node.

### 2.2.3. Node Importance Ranking

In the HNSN, it is necessary to identify the nodes with more significant influence on the network by ranking the nodes' importance and determining the order in which the nodes should be removed in the malicious attacks. Among the existing documentation, the significance of nodes is classified under various indicators. In some investigations, the importance of nodes is ranked according to an indicator. In contrast, others are based on comprehensive indicators. For instance, Yang proposed a weighted evaluation index using two indices (degree and betweenness centrality) [19]. Similarly, Chen also built a method

of ranking the importance of nodes by weighting different indexes (degree, betweenness centrality, and the oil volume) [20]. Although a variety of indicators are existed, based on existing literature, we find that: (1) topological structure of the network, such as degree and betweenness centrality, has been used as an index for ranking the significance of nodes in many books literature these two indices may accurately reflect network topological properties. (2) Indicators should be used to evaluate research objects. The purpose of this paper is to explore the safety of Hangzhou's NEV charging station. Therefore, the number of charge piles of each station node is one of the essential indicators.

Thus, we propose innovatively a new index of the significance of nodes, which combines the topological properties, node betweenness centrality, degree, and the minimum eigenvalues of the deleted Laplacian matrix of the network of each node and the number of recharging piles. We use the TOPSIS [21] method to construct this composite indicator.

We can construct a decision matrix as

$$
D = \begin{bmatrix}
k_1 & BC_1 & \lambda_{\min 1} & pile_1 \\
k_2 & BC_2 & \lambda_{\min 2} & pile_2 \\
\vdots & \vdots & \vdots & \vdots \\
k_n & BC_n & \lambda_{\min n} & pile_n
\end{bmatrix}
\tag{6}
$$

where $D$ is the minimum eigenvalues of the deleted Laplacian matrix, *pile* is the charging pile number of the charging station.

Next, the weight of the criteria is calculated by using the entropy method. Additionally, to eliminate magnitude and dimensional differences in values between indicators, the min-max normalization method is applied to normalize the values of these metrics between 0 and 1. The normalized value $\widetilde{x}$ is as follows:

$$
\widetilde{x_{ij}} = \frac{x_{ij} - \min\{x_{1j}, \cdots, x_{nj}\}}{\max\{x_{1j}, \cdots, x_{nj}\} - \min\{x_{1j}, \cdots, x_{nj}\}}, i = 1, \cdots, n; j = 1, \cdots, c
\tag{7}
$$

where $x_{ij}$ is the actual value of the node $v_i$ in the indicator of c. According to the reference, calculate the information entropy:

$$
p_{ij} = \frac{\widetilde{x_{ij}}}{\sqrt{\sum_{i=1}^{n} \widetilde{x_{ij}}}}, i = 1, \cdots, n; j = 1, \cdots, c
\tag{8}
$$

$$
en_j = -\frac{1}{\ln c} \sum_{i=1}^{n} p_{ij} \ln p_{ij}, i = 1, \cdots, n; j = 1, \cdots c
\tag{9}
$$

Calculating the weight of each indicator:

$$
G_j = 1 - en_j, j = 1, \cdots c
\tag{10}
$$

$$
W_j = \frac{G_j}{\sum\limits_{j=1}^{c} G_j}, j = 1, \cdots c
\tag{11}
$$

The comprehensive index of node $v_i$ importance:

$$
C_i = \frac{D_i^-}{D_i^+ + D_i^-}
\tag{12}
$$

where $D_i^+$ and $D_i^-$ are the node's Euclidean distances to the highest and the lowest importance.

## 3. Results and Discussion

### 3.1. Node Importance Ranking in HNSN

According to Equations (11) and (12), we can get the weight of the four indicators. Due to space limitations, only the node ranking of HNSN is shown here. The weights of the four indicators, including node betweenness centrality, degree, the minimum eigenvalue of the deleted Laplacian matrix, and the number of charging piles in HNSN, are 0.387, 0.163, 0.160, and 0.291, respectively. We ranked the importance of 186 nodes in HNSN according to the betweenness centrality of nodes, the degree of nodes, the minimum eigenvalues of the Laplacian matrix after node deletion, and the TOPSIS based on the entropy weight method. The ranking results of HNSN are presented in Figure 4.

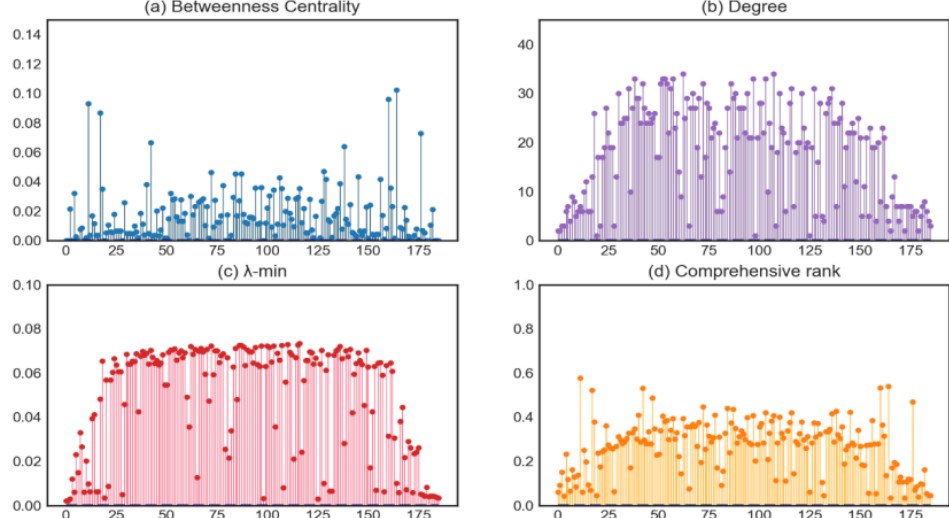

**Figure 4.** Ranking of key station nodes in HNSN. (**a**) The betweenness centrality of the station nodes. (**b**) The degree of the station nodes in HNSN. (**c**) The minimum eigenvalues of the deleted Laplacian matrix after removing the station nodes. (**d**) The comprehensive scores of the station nodes.

As shown in Figure 4a–d, only a few nodes in HNSN have relatively high betweenness centrality values, and most of the nodes have betweenness centrality values less than 0.04. What's more, nodes with higher betweenness centrality values also have comparatively higher comprehensive significance values. In Figure 4b, the degree of nodes in HNSN is relatively uniform, which indicates that the distribution of the NEV charging stations in Hangzhou is relatively uniform. Node N63 has the largest degree of 34, which means that node N63 has 34 neighbor nodes, that is to say, node N63 is connected to most of the nodes in its region. The network aggregation coefficient is 0.64, and it has small-world characteristics. Using the degree of the node and the minimum eigenvalue of the Laplacian matrix after deletion of the node solitary are inadequate for identifying key nodes in HNSN.

Only the top 30 nodes in HNSN are shown due to space constraints. As shown in Table 1, charging stations located in public facilities such as residential communities, transportation hubs, and scenic parks are ranked high, indicating the relatively high significance of residential communities. There are dense residential communities around the nodes of charging stations like V151, V47, V70, V88, V105, V106, and V86, so the population is heavily distributed and the flow of people is significant. Charging station nodes V161 and V120 are located in Hangzhou East Railway Station, so their comprehensive ranking is high as well. The charging station nodes N91, N72, N97, and N58 are located in West Lake, a famous scenic tourist area in Hangzhou. The charging station node V47 in Xiaoshan District is a large-scale commercial charging station operated by Teal. Thus, it has the largest number of charging piles. Therefore, people should pay more attention to the safety of charging stations built in transportation hubs, residential communities, and tourist attractions with high traffic flow.

**Table 1.** Ranking of key nodes in HNSN (top 30).

| ID | Coordinate | C-Pile Number | $\lambda_1(L_{N-1})$ | BC | Degree | C |
|---|---|---|---|---|---|---|
| v151 | (120.28089, 30.3239) | 10 | 0.05116 | 0.10764 | 14 | 0.54881 |
| v47 | (120.2478, 30.20251) | 133 | 0.18667 | 0.00228 | 40 | 0.43437 |
| v70 | (120.18848, 30.23318) | 26 | 0.30095 | 0.03858 | 72 | 0.40291 |
| v88 | (120.1639, 30.25835) | 12 | 0.29245 | 0.02328 | 68 | 0.39102 |
| v105 | (120.15646, 30.2741) | 30 | 0.29599 | 0.01865 | 69 | 0.37676 |
| v106 | (120.23602, 30.27511) | 2 | 0.1918 | 0.08272 | 36 | 0.3728 |
| v86 | (120.16863, 30.25399) | 2 | 0.29486 | 0.01854 | 66 | 0.3706 |
| v91 | (120.13026, 30.25961) | 20 | 0.27558 | 0.01387 | 61 | 0.36861 |
| v140 | (120.13556, 30.31095) | 8 | 0.28926 | 0.02038 | 65 | 0.36606 |
| v90 | (120.15864, 30.25932) | 2 | 0.29887 | 0.02329 | 71 | 0.3617 |
| v78 | (120.21018, 30.24084) | 24 | 0.26305 | 0.01265 | 57 | 0.34977 |
| v126 | (120.20942, 30.29852) | 30 | 0.20962 | 0.00636 | 40 | 0.34355 |
| v120 | (120.2129973, 30.29133) | 49 | 0.29152 | 0.007 | 51 | 0.34329 |
| v84 | (120.2107, 30.246624) | 16 | 0.27216 | 0.01889 | 59 | 0.34056 |
| v26 | (120.154, 30.17823) | 12 | 0.21648 | 0.02427 | 48 | 0.33844 |
| v62 | (120.17946, 30.22505) | 8 | 0.28035 | 0.01664 | 61 | 0.33801 |
| v133 | (120.12787, 30.30503) | 19 | 0.29184 | 0.01263 | 63 | 0.33638 |
| v129 | (120.24063, 30.30054) | 8 | 0.29378 | 0.02519 | 66 | 0.33259 |
| v77 | (120.2067, 30.24044) | 2 | 0.27683 | 0.01463 | 60 | 0.33175 |
| v72 | (120.16522, 30.23462) | 8 | 0.28083 | 0.02187 | 63 | 0.33036 |
| v94 | (120.1486, 30.26393) | 4 | 0.28341 | 0.01005 | 65 | 0.32645 |
| v101 | (120.16298, 30.26866) | 2 | 0.29184 | 0.01334 | 64 | 0.32522 |
| v97 | (120.13998, 30.26576) | 4 | 0.27947 | 0.00949 | 61 | 0.32333 |
| v67 | (120.17041, 30.22812) | 4 | 0.29151 | 0.01307 | 62 | 0.32291 |
| v58 | (120.15503, 30.21898) | 8 | 0.27505 | 0.01507 | 64 | 0.32078 |
| v64 | (120.16786, 30.22591) | 8 | 0.27527 | 0.01592 | 65 | 0.32078 |
| v59 | (120.13849, 30.22451) | 8 | 0.25075 | 0.01356 | 52 | 0.31923 |
| v141 | (120.15432, 30.31196) | 8 | 0.27638 | 0.01407 | 50 | 0.31836 |
| v52 | (120.20708, 30.21143) | 16 | 0.24605 | 0.00897 | 57 | 0.31535 |
| v134 | (120.142303, 30.3053) | 6 | 0.25195 | 0.01401 | 57 | 0.31387 |

*3.2. Node Attacks*

To study the robustness of WNSN and HNSN in the face of external attacks, we simulate random attacks and four devious attacks. The nodes' degree attack, the nodes' betweenness centrality attack, the deleted Laplacian matrix minimum eigenvalue attack, and the comprehensive rank attack are all performed in the decay sequence. We conduct 100 random attacks on HNSN and take the average value of those 100 random attacks to compare with the other four malicious on above.

We analyze the connectivity and efficiency of node attack strategies. Figures 5 and 6 indicate the robustness curves of the station network with five different attack methods; the curves show the relative giant connected component size and efficiency of the network decreases under attacks. From Figures 5 and 6, we can see that the efficiency of the charging station network is poor in both WNSN and HNSN. HNSN has better performance in efficiency. Overall, as shown in Figures 5 and 6, the performance of WNSN and HNSN in the face of attacks both are poor, whether in responding to malicious attacks or random attacks. What is more, whether it is the connectivity or the efficiency of the HNSN, the betweenness centrality attacking index causes substantial damage. The strategy can attack crucial nodes at the very beginning, so network connectivity and efficiency can quickly collapse. In Section 3.1, we know that the distance between NEV charging stations in Hangzhou is relatively close, the distribution is relatively uniform, and the degree value of each node of HNSN is not very different. Therefore, in terms of destroying the connectivity of the network, the performance of the degree attack is not as destructive as that of the random attack. However, degree attack is more destructive in reducing network efficiency than random attacks. Similarly, the minimum eigenvalue of the Laplacian matrix after node deletion is less destructive than the random attack. However, it has been proved by

experiments that this method has a more obvious attack effect on the scale-free network, and has no significant effect on this kind of relatively uniform network attack. Since the method of degree attack and the minimum eigenvalue attack of the Laplacian matrix after node removal with no obvious attack effect are considered together with the node betweenness centrality with obvious attack effect, the comprehensive attack method has a poor attack effect. Similarly, we found that by combining the three malicious attack methods and the number of charging piles per charging station node, the connectivity curve of the network decreased less. Moreover, compared with the other three malicious attack strategies, HNSN's network has the slowest decline in network efficiency under this malicious attack. This means that if we take these factors into account, we can relatively improve the robustness of HNSN against malicious attacks.

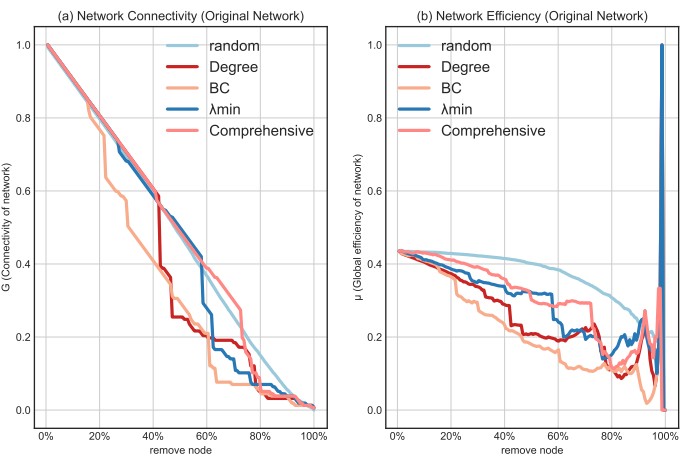

**Figure 5.** Random attack and malicious attack effect in WHSN. (**a**) The change in original network connectivity after deleting nodes in sequence according to the 5 methods. (**b**) The change in original network efficiency after deleting nodes in sequence according to the 5 methods.

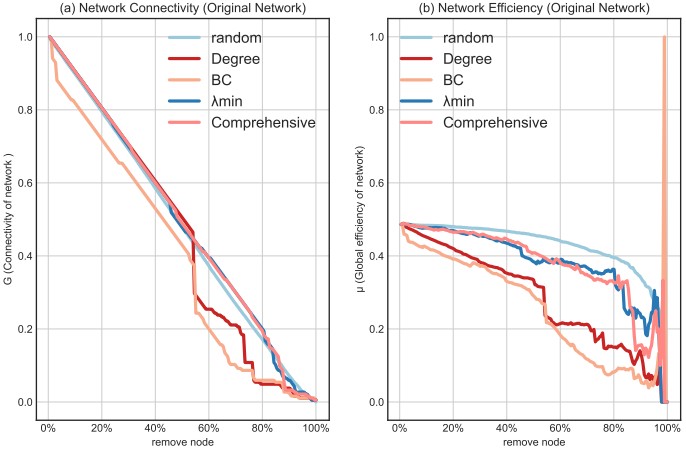

**Figure 6.** Random attack and malicious attack effect in HNSN. (**a**) The change in original network connectivity after deleting nodes in sequence according to the 5 methods. (**b**) The change in original network efficiency after deleting nodes in sequence according to the 5 methods.

## 4. Construct a Virtual HNSN by Adding Nodes

Some scholars have believed that districts with different land usage, such as residential areas, business districts, and along roads, will generate different charging needs [22]. Through the analysis of drivers' daily activities, González et al. found that human activities can influence the charging demand, generating a large amount of charging demand

in residential places, followed by workplaces, and finally, shopping activities are important for demand forecasting [23]. Specifically, the virtual network is built with the following conditions:

- Affordable:

  At present, the number of existing NEVs is not enough to support the large-scale laying of charging piles, so it is not appropriate to build too many charging stations and charging piles.

- Appropriate:

  NEV charging stations need to be built in places with dense traffic and convenient transportation such as transportation hubs, shopping malls, and residential quarters. Additionally, the new nodes need to be able to connect to the unconnected original in the original network.

  The information on new energy charging piles is shown in Table 2. We take the average number of charging piles of 186 energy vehicle charging stations in Hangzhou as the number of charging piles for new charging stations. Numbered lists can be added as follows:

**Table 2.** The information of the new charging stations.

| New Station ID | Longitude | Latitude | Pile Number |
|---|---|---|---|
| N-S1 | 120.1104 | 30.1442 | 13 |
| N-S2 | 120.078 | 30.1685 | 13 |
| N-S3 | 120.1103 | 30.1179 | 13 |
| N-S4 | 120.0876 | 30.1701 | 13 |
| N-S5 | 120.1629 | 30.1332 | 13 |
| N-S6 | 120.1293 | 30.1292 | 13 |
| N-S7 | 120.1414 | 30.1526 | 13 |
| N-S8 | 120.2958 | 30.2778 | 13 |
| N-S9 | 120.2875 | 30.336 | 13 |
| N-S10 | 120.2334 | 30.373 | 13 |
| N-S11 | 120.2798 | 30.3502 | 13 |
| N-S12 | 120.3142 | 30.3211 | 13 |
| N-S13 | 120.2222 | 30.3652 | 13 |
| N-S14 | 120.2487 | 30.1519 | 13 |
| N-S15 | 120.2909 | 30.2584 | 13 |
| N-S16 | 120.2626 | 30.3116 | 13 |
| N-S17 | 120.2791 | 30.1559 | 13 |
| N-S18 | 120.239 | 30.1666 | 13 |
| N-S19 | 120.2162 | 30.3136 | 13 |

We also executed the above five attack strategies on the built virtual HNSN by adding new station nodes. The robustness curves of the virtue HNSN are shown in Figure 7. Comparing Figures 6 and 7, we can find that the robustness of HNSN to external attacks can be enhanced by adding new station nodes. When the proportion of the attacked nodes is less than 40%, there is little difference between malicious and random attacks in reducing the connectivity of the network. That is to say, when the proportion of attacked nodes is less than 40%, under the most malicious attacking method of betweenness centrality, the virtual HNSN will slow down the rate of network connectivity decline. This means the approach of adding new station nodes can improve the connectivity of the original network. Comparing Figures 6 and 7, it can be seen that when 40% of the nodes in the network are attacked, the efficiency of the original network drops to around 0.2, while in the virtual network, the efficiency of the network drops to about 0.25, so the robustness of the network is improved. Even with the increase in nodes, the comprehensive attack strategy still has the worst attack effect.

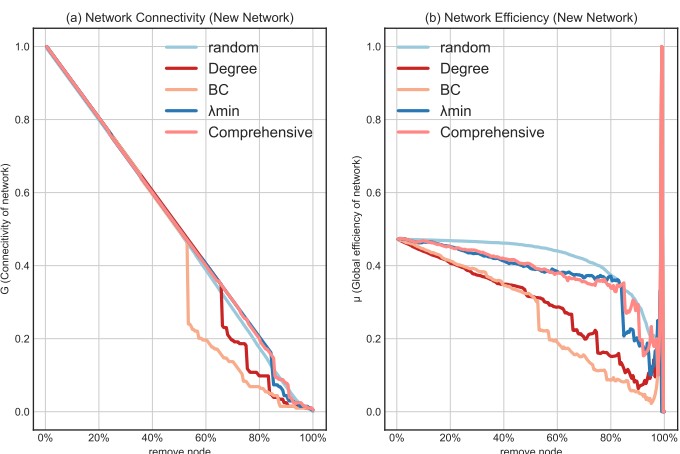

**Figure 7.** Virtual network random attack and malicious attack effect. (**a**) The change in virtual network connectivity after deleting nodes in sequence according to the 5 methods. (**b**) The change in virtual network efficiency after deleting nodes in sequence according to the 5 methods.

## 5. Conclusions and Suggestions

Although no network is entirely immune to risks, a relatively robust network can remain robust in uncertain times. Firstly, the HNSN is built based on the number of charging piles and the geographical location of the NEV charging stations in Hangzhou. The next step is to perform random attacks and malicious attacks on HNSN and analyze the network's resilience to each attack method. Finally, a virtual network is constructed to verify whether HNSN is more robust. The evidence from this study points towards the idea that making the HNSN more robust is a pressing need to secure energy. Our main findings are as follows:

1. The robustness of the HNSN is poor. The NEV charging stations in Hangzhou are distributed evenly, but the whole charging station network is inefficient;
2. Moreover, the betweenness centrality attack can cause more damage to HNSN. Since we built the HNSN based on distance, and the charging stations in Hangzhou are more evenly distributed, the degree attack cannot identify key nodes well;
3. New station nodes can enhance the robustness of HNSN. In particular, after adding new nodes, the performance of the betweenness centrality attack strategy in decreasing network connectivity is reduced.

As the country's environmental governance continues to rise, restrictions on the use of odd and even fuel vehicles have resulted in travel restrictions for car owners. Public awareness of environmental protection and green commuting has increased year over year, and renewal and demand for NEVs are also increasing year over year, the base of new demand and renewal demand continues to grow annually. In light of the above research, we make the following suggestions:

1. Focus on charging stations built in transportation hubs, large shopping malls, community gathering places, and tourist attractions with a large number of passengers. Standardize daily charging station management measures to prevent hazardous events that affect the safety of charging station facilities. More charging piles can be placed in the neighbor station nodes of the station nodes with large degrees;
2. Actively build NEV charging stations. Considering the constructed network of NEV charging stations in Hangzhou, the current urban charging station network is relatively evenly distributed, but the overall efficiency of the network is low and the robustness is poor;

3. Improve battery technology. The land for the construction of NEV charging stations is a non-renewable resource, so it is not feasible to build new charging stations blindly. We need to accelerate the promotion of battery technology to improve battery life.

The most important limitation lies in our model being much simpler than the actual conditions. In the future, we will try to combine the charging station network and the traffic network, considering the traffic flow, to test the maximum charging capacity of this network model that combines the two types of networks, in order to provide new opinions on charging station planning.

**Author Contributions:** Conceptualization, P.Z. and J.C.; methodology, P.Z. and J.C.; software, P.Z.; validation, P.Z.; formal analysis, P.Z., L.T. and J.C.; investigation, P.Z. and L.Y.; resources, P.Z., L.T. and J.C.; data curation, P.Z.; writing—original draft preparation, P.Z., L.T., L.Y. and J.C.; writing—review and editing, P.Z., L.T. and J.C.; visualization, P.Z. and J.C.; supervision, J.C.; project administration, L.T. and J.C. All authors have read and agreed to the published version of the manuscript.

**Funding:** This research received no external funding.

**Institutional Review Board Statement:** Not applicable.

**Informed Consent Statement:** Not applicable.

**Data Availability Statement:** Not applicable.

**Acknowledgments:** We are grateful for the help of the mobile application Chongdianba.

**Conflicts of Interest:** The authors declare no conflict of interest.

## Abbreviations

The following abbreviations are used in this manuscript:

| | |
|---|---|
| NEV | New Energy Vehicle |
| WNSN | Wuhan New Energy Stations Network |
| HNSN | Hangzhou New Energy Stations Network |
| MCDM | Multi-Criteria Decision-Making Methods |

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
