# Peer review of "Layout Evaluation of New Energy Vehicle Charging Stations: A Perspective Using the Complex Network Robustness Theory"

_wevj, doi:10.3390/wevj13070127_

Round 1

Reviewer 1 Report

1. Improve the abstract in more precise way.

2. Research contributions to be included in introduction.

3. Quality of the figures can be improved.

4. Equ 5 recheck

The following paper is missed in introduction which it tell about the optimization techniques of all charging methods.

Rajanand Patnaik Narasipuram, Subbarao Mopidevi, A technological overview & design considerations for developing electric vehicle charging stations, Journal of Energy Storage, Volume 43, 2021, 103225.

Author Response

Response to Reviewer 1 Comments

The co-authors and I would like to thank you for the time and effort spent in reviewing the manuscript. We have carefully revised the language based on the latest file uploaded on June 20th. Based on the reviewers' comments, we added a new city case for analysis, which greatly enriched the content of the article. We have added the case of Wuhan and analyzed it detailly in sections 2.2 and 3.2. In addition, we have made major revisions to the introduction and literature review, which mainly on page 1-3. This reference that you provided to help us add background knowledge has been added to reference 5, which you can find on page 1, line 23, and page 13, lines 397-398. This revision is highlighted in red in the text. Also, we have uploaded the revised manuscript file. Our point-by-point responses are detailed below.

Point 1: Improve the abstract in more precise way. 

Response 1: We are grateful for the suggestion. To be more clearly and in accordance with the reviewer’s concerns, we have revised the abstract, outlined the methods and models used in this paper. More details ware added on page 1, lines 1 -10.

Point 2: Research contributions to be included in introduction.

Response 2: We deeply appreciate the reviewer’s suggestion. According to the reviewer’s comment, we revised the introduction and added the contributions of this paper to the introduction, which is modified on page 3, lines 99-108.

Point 3: Quality of the figures can be improved.

Response 3: Thank you for your suggestion. As suggested by reviewer, we have re-plot Figure 2 and Figure 3 based on the data, which are modified on the bottom of page 4.

Point 4: Equ 5 recheck.

Response 4: We are extremely grateful to reviewer for pointing out this problem. As suggested by reviewer, we have modified Equation 5, which is modified on page 6, lines 198. 

Thank you for your valuable and thoughtful comments.

Reviewer 2 Report

The authors have executed a very effective study by implementing five attack strategies on the built virtual HNSN by adding 272 new station nodes.

The comparison between  Figure 6 and Figure 7 clearly manifests that 40% of the nodes in the network are attacked, the efficiency of the original network drops to around 0.2, while  in the virtual network, the efficiency of the network drops to about 0.25, thereby improving the robustness of the network. The evidence from this study points towards the  idea that making the HNSN more robust is a pressing need to secure energy is the need of the hour.

The authors have done a great job. I just have one suggestion. In figure 6 and figure 7, please mention what is G and U (Myu) in the axis of the plot. I am recommending the paper towards acceptance with minor revision.

Author Response

Response to Reviewer 2 Comments

The co-authors and I would like to thank you for the time and effort spent in reviewing the manuscript. We have carefully revised the language based on the latest file uploaded on June 20th. Based on the reviewers' comments, we added a new city case for analysis, which greatly enriched the content of the article. We have added the case of Wuhan and analyzed it detailly in sections 2.2 and 3.2. In addition, we have made major revisions to the introduction and literature review, which mainly on page 1-3. This revision is highlighted in red in the text. Also, we have uploaded the revised manuscript file. Our point-by-point responses are detailed below.

Point 1: Mention what is G and U (Myu) in the axis of the plot.

Response 1: We are grateful for the suggestion. We reran the program, and labelled the meaning of the G and μ representations on the vertical axes in Figures 5, 6 and 7. For details, please see Figure 5, Figure 6 on page 10, and Figure 7 on page 12.

Reviewer 3 Report

The topic of paper titled “Layout evaluation of New-Energy vehicle charging stations: A perspective using the complex network robustness theory “ is quite interesting and mostly well presented, but there are also some important shortcomings. I have some questions and suggestions for improvements. I am asking the authors to comment on them:

1. Reading the article you get the impression that it is a case study of one city. This should be more generalized and presented a universal analysis or comparison with another case.

2. In the introduction, a general transport background should be introduced. The reference should also be made to electric public transport vehicles and Electric Multiple Units.

3. Were all the figures made by yourself? Any copyright, especially figure 3, are not needed?

4. What are the further development plans for the conducted research? What are the advantages and disadvantages of the presented method?

5. I suggest also taking into account other foreign experience in this subject, not only in China.

I have also one suggestion regarding to text formatting:

6. The readability of the charts should be improved.

7. In some places, text formatting (line spacing under figure captions) needs to be improved.

I think the topic of research has a potential but it should be clearly indicated and emphasized in the content of the paper. It is also necessary to outline the background better, develop the analysis further (e.g. add one more case) and refer to other experiences. I would also like to know what is the further research plan of the authors on this paper.

Author Response

Response to Reviewer 3 Comments

The co-authors and I would like to thank you for the time and effort spent in reviewing the manuscript. We have carefully revised the language based on the latest file uploaded on June 20th. Based on the reviewers' comments, we added a new city case for analysis, which greatly enriched the content of the article. We have added the case of Wuhan and analyzed it detailly in sections 2.2 and 3.2. In addition, we have made major revisions to the introduction and literature review, which mainly on page 1-3. This revision is highlighted in red in the text. Also, we have uploaded the revised manuscript file. Our point-by-point responses are detailed below.

Point 1: Reading the article you get the impression that it is a case study of one city. This should be more generalized and presented a universal analysis or comparison with another case.

Response 1: We are extremely grateful to reviewer for pointing out this problem. The analysis of one city is indeed inadequate. To be more clearly and in accordance with the reviewer’s concerns, we have added a new city case for analysis. Due to the availability of data, it is difficult for us to obtain data from foreign cities, so this paper supplements the case of new energy vehicle charging facilities in Wuhan of China. More details ware added on page 4, section 2.2 and page 9, section 3.2.

Point 2: In the introduction, a general transport background should be introduced. The reference should also be made to electric public transport vehicles and Electric Multiple Units.

Response 2: Thanks for your suggestions. We agree with the comment and re-wrote the introduction and literature review, supplied some transport background and some reference about electric vehicles and electric vehicles infrastructure. More details ware added on page 1-3.

Point 3: Were all the figures made by yourself? Any copyright, especially figure 3, are not needed?

Response 3: Thank you very much for mentioning the copyright. In the previous version, all the pictures were drawn by us. For the Figure 3 you mentioned, we used Python to call the base map provided by OpenStreetMap and mapped our data on the map for a secondary creation. OpenStreetMap is a map of the world, constructed by people like you, and is freely available under an open license. It mentions that we are free to copy, distribute, transmit and adapt our data, as long as we credit OpenStreetMap and its contributors. To avoid copyright disputes, we have modified the previous version of Figure 3. We have used the administrative area data published by the National Catalogue Service For Geographic Information, and according to its copyright requirements, "Reproduction or quotation of all the contents of this website must indicate the words ‘transferred from (or quoted from) the National Catalogue Service For Geographic Information ‘, and indicate our website URL www. webmap.cn", we have marked it, please refer to page 4, lines 137-138 for details.

Point 4: What are the further development plans for the conducted research? What are the advantages and disadvantages of the presented method?

Response 4: Urban charging facilities are modeled as a network system using complex network theory to better identify their overall effects as well as the interactions between individuals within the system. However, real-world charging facility systems are more complex, and our model is a simplified system that does not consider the role of charging modes and the influence of the traffic road network. In the future, we would like to continue to use complex network theory to embed the charging station network into the traffic road network, taking into account the influence of charging modes as well as traffic flow and other factors.

Point 5: I suggest also taking into account other foreign experience in this subject, not only in China.

Response 5: We deeply appreciate the reviewer’s suggestion. It is partial to consider only the experience of China. So we have append some foreign literature about planning of charging stations, such as the USA, Japan, and South Korean. More details are in page 1, lines 17-21.

Point 6: The readability of the charts should be improved. In some places, text formatting (line spacing under figure captions) needs to be improved.

Response 6: We are grateful for the suggestion. The typographical issues have been adjusted in the latest version.

Point 7: The further research plan

Response 7: Thank you very much for your interest in our future research programs. We plan to embed the charging facility network into the traffic road network in the future and analyze the maximum capacity of urban charging resources at the moment, taking into account factors such as traffic flow charging modes and queuing time. However, the idea is not yet mature and it is difficult to obtain data, so I hope to get your advice. About our further research plan, I have added in the page 13, lines 369-372.

Round 2

Reviewer 3 Report

The authors of the paper took into account the comments and provided comprehensive answers to the questions asked. In my opinion, the article can be accept in present form.